# Synergy of Chemo- and Photodynamic Therapies with C_60_ Fullerene-Doxorubicin Nanocomplex

**DOI:** 10.3390/nano9111540

**Published:** 2019-10-30

**Authors:** Anna Grebinyk, Svitlana Prylutska, Oksana Chepurna, Sergii Grebinyk, Yuriy Prylutskyy, Uwe Ritter, Tymish Y. Ohulchanskyy, Olga Matyshevska, Thomas Dandekar, Marcus Frohme

**Affiliations:** 1Division Molecular Biotechnology and Functional Genomics, Technical University of Applied Sciences Wildau, Hochschulring 1, 15745 Wildau, Germany; grebinyk@th-wildau.de (A.G.); sgrebinyk@th-wildau.de (S.G.); 2Department of Bioinformatics, Biocenter, University of Würzburg, Am Hubland, 97074 Würzburg, Germany; dandekar@biozentrum.uni-wuerzburg.de; 3Taras Shevchenko National University of Kyiv, Volodymyrska 64, 01601 Kyiv, Ukraine; psvit_1977@ukr.net (S.P.); prylut@ukr.net (Y.P.); 4Key Laboratory of Optoelectronic Devices and Systems of Ministry of Education and Guangdong Province, College of Physics and Optoelectronic Engineering, Shenzhen University, Shenzhen 518060, China; ok.chepurna@gmail.com (O.C.); tyo@szu.edu.cn (T.Y.O.); 5Institute of Chemistry and Biotechnology, University of Technology Ilmenau, Weimarer Straße 25 (Curiebau), 98693 Ilmenau, Germany; uwe.ritter@tu-ilmenau.de; 6Palladin Institute of Biochemistry, NAS of Ukraine, Leontovicha Str. 9, 01030 Kyiv, Ukraine; matysh@yahoo.com

**Keywords:** photodynamic chemotherapy, synergistic effect, C_60_ fullerene, Doxorubicin, nanocomplex, leukemic cells, apoptosis

## Abstract

A nanosized drug complex was explored to improve the efficiency of cancer chemotherapy, complementing it with nanodelivery and photodynamic therapy. For this, nanomolar amounts of a non-covalent nanocomplex of Doxorubicin (Dox) with carbon nanoparticle C_60_ fullerene (C_60_) were applied in 1:1 and 2:1 molar ratio, exploiting C_60_ both as a drug-carrier and as a photosensitizer. The fluorescence microscopy analysis of human leukemic CCRF-CEM cells, in vitro cancer model, treated with nanocomplexes showed Dox’s nuclear and C_60_’s extranuclear localization. It gave an opportunity to realize a double hit strategy against cancer cells based on Dox’s antiproliferative activity and C_60_’s photoinduced pro-oxidant activity. When cells were treated with 2:1 C_60_-Dox and irradiated at 405 nm the high cytotoxicity of photo-irradiated C_60_-Dox enabled a nanomolar concentration of Dox and C_60_ to efficiently kill cancer cells in vitro. The high pro-oxidant and pro-apoptotic efficiency decreased IC_50_ 16, 9 and 7 × 10^3^-fold, if compared with the action of Dox, non-irradiated nanocomplex, and C_60_’s photodynamic effect, correspondingly. Hereafter, a strong synergy of therapy arising from the combination of C_60_-mediated Dox delivery and C_60_ photoexcitation was revealed. Our data indicate that a combination of chemo- and photodynamic therapies with C_60_-Dox nanoformulation provides a promising synergetic approach for cancer treatment.

## 1. Introduction

Chemotherapy (CT) as one of the conventional cancer therapies aims to slow down the growth of cancer cells that evolved in a fast proliferation [1]. The frontline anthracycline chemotherapeutic drug Doxorubicin (here abbreviated consistently Dox) intercalates into nuclear DNA initiating the suppression of topoisomerase activity as well as of DNA transcription, replication, and repair [2,3,4]. Dox also affects the redox balance inducing reactive oxygen species (ROS) generation through metal chelation and flavoprotein reductase-associated redox cycling [4,5]. The extended ROS generation causes serious cardiotoxicity owing to the high content of mitochondria in cardiomyocytes [2,3,6] and limits the drug’s clinical application.

The multimodal combination of therapies with distinct anticancer mechanisms offers potential advantages and enhanced efficiency compared to monotherapy approaches [7,8]. Photodynamic chemotherapy exploits two anticancer agents—a photosensitizing molecule and a chemotherapeutic drug [9,10,11,12,13,14,15,16]. The former is harmless itself but on illumination with visible light in the presence of oxygen induces cell death through ROS-mediated compact apoptosis [17,18]. Once combined, photodynamic therapy (PDT) and CT confront cancer cells with two different “swords”, resulting in a stronger therapeutic potential in comparison with the corresponding therapies apart or their theoretical sum. The advantageous synergistic effect is primarily attributed to the heterogeneity of cancer cell resistance to each of the monotherapies and finally aims to apply lower clinical dosage of the chemotherapeutics [8].

Photosensitizers and Dox are applied either separately as a co-treatment [19,20] or bound on the nanocarrier platform [9,10,11,12,13,14,15,16]. The enhanced efficiency of Dox included in nanoplatforms together with such photosenitizers as chlorin 6 [11,13,14], phthalocyanines [12,15,21], indocyanine green [22], merocyanine [9], methylene blue [10], and perfluorocarbon [16] was recently reported. The carbon nanostructure C_60_ fullerene [23] (here abbreviated consistently C_60_) has attracted attention as a photosensitizer [1,24,25,26,27] due to its unique physicochemical behavior, high quantum yield of ROS production [28], photostability, low photobleaching [25] as well as predominant mitochondrial localization [29,30,31,32]. Pristine C_60_ stable colloid solution with a negligible toxicity against normal cells [33,34] was explored for PDT [26,28,31,35,36,37,38]. A pronounced pro-apoptotic effect was detected in leukemic cells treated with pristine ≤20 µM C_60_ and irradiated with UV-Vis light in the range of 320 – 600 nm [26,35,36,37,39,40]. A continuous intensification of ROS production and inhibition of glutathione-dependent antioxidant system testified a subsequent intense induction of oxidative stress [40]. As a result, store-operated Ca^2+^ entry and cytochrome c release from mitochondria [37] induced Ca^2+^-dependent apoptosis of leukemic cells [39].

The combination of the C_60_ pro-oxidant properties [26,31,36,37] and its drug delivery capability [24,41,42] makes this nanostructure attractive for cancer photodynamic chemotherapy (Figure 1). The purpose of this study was to assess the toxic effects of the non-covalent C_60_-Dox nanocomplex in combination with light irradiation (405 nm emission from high power single chip LED) on human leukemic CCRF-CEM cells. The nanocomplex was designed in two nanomolar ratios of C_60_ to Dox (1:1 and 2:1) in order to investigate whether C_60_’s concentration affected Dox efficacy. Firstly, the intracellular localization of the nanocomplexes in leukemic cells was estimated with fluorescence microscopy following immunofluorescence staining. Then, the leukemic cells’ viability was studied upon treatment with the nanocomplexes and 405 nm LED light irradiation. The anticancer pro-apoptotic potential of the combinative treatment was assessed by evaluation of intracellular ROS production, caspase 3/7 activity, ATP level, and phosphatidylserine translocation in the plasma membrane in leukemic cells.

## 2. Materials and Methods 

### 2.1. Chemicals

Roswell Park Memorial Institute (RPMI) 1640 liquid medium, phosphate buffered saline (PBS), fetal bovine serum (FBS), penicillin/streptomycin, and L-glutamin were obtained from Biochrom (Berlin, Germany). Poly-D-lysine hydrobromide, triton X100, bovine serum albumin (BSA), 4′,6-diamidine-2′-phenylindole dihydrochloride (DAPI), glycerol, 3-(4,5-dimethylthiazol-2-yl)-2,5-diphenyl tetrazolium bromide (MTT), and Dox were obtained from Sigma-Aldrich Co. (St-Louis, MI, USA). Paraformaldehyde, dimethylsulfoxide (DMSO), sodium chloride, acetonitrile, formic acid, and trypan blue from Carl Roth GmbH + Co. KG (Karlsruhe, Germany) were used. 

### 2.2. C_60_ and C_60_-Dox Nanocomplex

The pristine C_60_ aqueous colloid solution was prepared by C_60_ transfer from toluene to water using continuous ultrasound sonication as described by Ritter et al. [43]. The obtained aqueous colloid solution of C_60_ was characterized by 0.2 mM C_60_ concentration, 99% purity, stability, and homogeneity; the average size of nanoparticles was 100 nm [43,44].

Dox was dissolved in water at 18.5 mM initial concentration. 

C_60_-Dox nanocomplexes were created according to the following protocol. Briefly, Dox solution was mixed with C_60_ colloid solution in 1:1 or 2:1 molar ratio of the components. The mixture was treated in the ultrasonic disperser for 30 min, stirred for 24 h at room temperature (RT) and centrifuged at 14,000 g for 15 min with the use of centrifuge filters Amicon Ultra-0.5 3 K (Sigma-Aldrich Co., St-Louis, MI, USA) for sample purification. The stability and concentration of the nanocomplexes were controlled with dynamic light scattering and high-performance liquid chromatography–mass spectrometry. The stability (ζ-potential value) and size distribution (hydrodynamic diameter) of nanocomplexes as described in ref. [45] was systematically checked and shown to be practically unchanged after 6 months of storage in physiological saline solution. The concentration of C_60_ and Dox in the stock solution of 1:1 nanocomplex was 100 µM. The concentration of C_60_ and Dox in the stock solution of 2:1 nanocomplex was 200 µM (198.9 µM) and 100 µM, correspondingly.

Measuring the value of the translational diffusion coefficient as a function of C_60_ concentration at constant Dox concentration was performed in ref. [46]. The diffusion curve displays very distinct changes at small C_60_ concentrations and reaches a plateau for higher concentrations. It allows the conclusion that the binding of Dox molecules mainly occurs with C_60_ nanoclusters in aqueous solution. Indirectly this conclusion is confirmed by the atomic force microscopy data [46]; most likely C_60_-Dox nanocomplexes stabilized by the Dox-induced attenuation of electrostatic repulsion between C_60_.

Finally, the maximum number of ligand molecules (*N*) that can be bound with C_60_ in aqueous solution may be estimated independently with the help of proposed general up-scaled model [47] as well as molecular modeling. The following value was obtained for Dox: *N* = 3 [48,49]. The calculated equilibrium hetero-complexation constant was equal to *K_L_* ≈ 60,000 M^−1^ [47].

The content of 1:1 and 2:1 C_60_-Dox nanocomplexes after incubation in RPMI medium for 24 h was assessed to account for 81.50% ± 5.03% and 83.83% ± 5.47%, correspondingly, of the respective 0 h control (Figure 2). For this, C_60_-Dox nanocomplexes were incubated in RPMI up to 24 h under the identical conditions adopted from cell-based experiments (450 nM, 2 mL, 37 °C). For sample purification from a released free drug, 500 µL of each sample was filtered with the centrifugal filter devices Amicon Ultra-0.5 3 K (Sigma-Aldrich Co., St-Louis, MI, USA) according to the manufacturer’s instructions: 14,000 g, 15 min for filtration; 1000 g, 2 min for recovery (reverse spin upside down in a new centrifuge tube). The content of the filter device was subjected to optical analysis. C_60_-Dox nanocomplexes samples (50 μL) were placed into 384-well plate Sarstedt and fluorescence intensities were measured with a multimode microplate spectrometer Tecan Infinite M200 Pro (Tecan Infinite M200 Pro, Männedorf, Switzerland) at the following parameters: λ_ex_ = 470 nm, λ_em_ = 595 nm, number of flashes per well—25, integration time—20 µs. The obtained data were normalized with the RPMI control and expressed as percentage of the respective control sample, analyzed at 0 h.

The used concentrations of the C_60_-Dox nanocomplexes for cells treatment were 50, 150, and 450 nM, presented according to Dox concentrations in order to compare the effect of the nanocomplexes with the effect of the free drug.

### 2.3. LED Light Source for Photodynamic Therapy

For cell treatment in well plates a LED-based system was developed (Figure 3). The light source system consists of control and irradiation units. Taking our requirements from our recent experiments into account [31] we set up the irradiation unit with a high power single chip 405 nm LED VL400-EMITTER (Roithner Lasertechnik GmbH, Vienna, Austria) on a cylindrical heat sink (Figure 3a). The cascade of lens was designed to ensure high irradiation power density and even illumination over the well (Figure 3a). For the development of the optical cascade we applied an aspherical lens for reducing the divergence angle of the beam (D = 13.0 mm, h = 7.1 mm from Cree Inc., Durham, North Carolina, USA), which allowed all light to be focused to a second spherical lens with 35° (50% int) angle (D = 16.4 mm, h = 5.0 mm from Cree Inc., Durham, NC, USA) for increasing the radiation density. The diameter of the collimated beam was determined by the distance between the two lenses. The light system provides the same power density at any point of irradiation. The maximum diameter of the beam was 35 mm and the minimum 25 mm with 130 mW power. The light fluence was used at either 5 or 10 J/cm^2^ for comparison of cell treatment effect. The mounting carcass was built in SOLIDWorks from Dassault Systems (Vélizy-Villacoublay, France) (Figure 3b) and 3D-printed at Ultimaker 2+ (Utrecht, The Netherlands) (Figure 3c). The final light system was constructed with a metal turning and assembled at the Fotonika Plus Co. (Cherkasy, Ukraine) (Figure 3d).

### 2.4. Cell Culture 

The human cancer T-cell line CCRF-CEM (ACC 240) of leucosis origin was purchased from the Leibniz Institute DSMZ-German Collection of Microorganisms and Cell Cultures (Deutsche Sammlung von Mikroorganismen und Zellkulturen, Braunschweig, Germany).

Cells were maintained in 5 mL RPMI 1640 medium supplemented with 10% FBS, 1% Penicillin/Streptomycin and 2 mM Glutamine, using 25 cm^2^ flasks at a 37 °C with 5% CO_2_ in a humidified incubator Binder (Tuttlingen, Germany). The number of the viable cells was counted with the use of Roche Cedex XS Analyzer (Basel, Switzerland) after staining with 0.1% trypan blue.

### 2.5. Immunofluorescence Staining of C_60_


CCRF-CEM cells (2 × 10^5^/2 mL) were seeded in 6-well plates (Sarstedt, Nümbrecht, Germany) on cover slips (Carl Roth, Karlsruhe, Germany), previously coated with poly-D-lysine and incubated for 24 h. Cells were treated with free Dox, C_60_ or C_60_-Dox nanocomplexes in a 450 nM Dox equivalent concentration for a further 24 h. Then the cells were washed with PBS and fixed with 4% paraformaldehyde for 15 min at room temperature (RT) in the dark. After washing with PBS, the cells were permeabilized with 0.2% triton X100 for 10 min at RT and washed again with PBS. Blocking was performed using 10% BSA for 20 min followed by washing in PBS. The primary C_60_-mouse monoclonal IgG antibody bound to bovine thyreoglobulin (dilution ratio of 1:30 in PBS/1.5% BSA, 1-10F-A8 Santa Cruz Biotechnology Inc., Santa Cruz, California, USA) was added to the medium and CCRF-CEM cells were incubated overnight at 4 °C in a humidified chamber. Then CCRF-CEM cells were incubated for 3 h at RT with a fluorescein isothiocyanate- (FITC) labeled polyclonal rabbit-anti-mouse IgG antibody (dilution ratio of 1:200 in PBS/1.5% BSA, F7506 Sigma-Aldrich Co., St-Louis, MI, USA). Slides were washed with PBS for 15 min three times. Coverslips were rinsed with dH_2_O, incubated for 2 h in the dark with the nucleus staining/antifade solution (0.6 µM DAPI, 90 mM p-phenylenediamine in glycerol/PBS) and sealed with slides.

### 2.6. Fluorescence Microscopy

Images of CCRF-CEM cells stained with DAPI, Dox, and FITC-labeled antibodies against C_60_, were viewed with a fluorescence microscope Keyence BZ-9000 BIOREVO (Osaka, Japan). The microscope was equipped with blue (λ_ex_ = 377 nm, λ_em_ = 447 nm), green (λ_ex_ = 472 nm, λ_em_ = 520 nm), and red (λ_ex_ = 543 nm, λ_em_ = 593 nm) filters. The acquisition Keyence BZ-II Viewer Software (Osaka, Japan) was used. The merged images were processed with the Keyence BZ-II Analyzer Software (Osaka, Japan).

### 2.7. Photodynamic Therapy In Vitro and Cell Viability Assay

CCRF-CEM cells (10^4^/well) were cultured in 96-well cell culture plates (Sarstedt, Nümbrecht, Germany) for 24 h. The cell culture medium was replaced by 1% FBS drug-contained medium. Cells were incubated in the presence of 50, 100, and 450 nM Dox or C_60_-Dox nanocomplexes in Dox equivalent concentrations. After 24 h incubation cells were washed with PBS and irradiated with the developed 405 nm high power single chip LED light source (108.3 mW/cm^2^, 5 or 10 J/cm^2^). PBS was replaced with the fresh medium immediately after irradiation. Control cells were incubated without any treatment or light irradiation. After 24 h incubation, cell viability was determined with MTT reduction assay [50]. 10 μL of MTT solution (5 mg/mL in PBS) was added to each well and cells were incubated for 2 h at 37 °C. The culture medium was then replaced with 100 μL of DMSO and in 15 min diformazan formation was determined by measuring the absorption at 570 nm with a microplate reader (Tecan Infinite M200 Pro, Männedorf, Switzerland). Curve fitting and calculation of the half-maximal inhibitory concentration (IC_50_) values were done using the specialized software GraphPad Prism 7 (GraphPad Software Inc., San Diego, CA, USA). Briefly, individual concentration-effect curves were generated by fitting the logarithm of the tested compound concentration versus the corresponding normalized percent of cell viability using nonlinear regression.

### 2.8. Intracellular Reacrive Oxygen Species Generation

To determine ROS production 2,7-dichlorofluorescin diacetate (DCFH-DA, Sigma-Aldrich Co., St-Louis, Missouri, USA) was applied. A 5 mM stock solution of DCFH-DA was prepared in DMSO, stored at −20 °C and diluted with PBS immediately before use. CCRF-CEM cells were seeded into 96-well plates (10^4^ cells/well) and incubated for 24 h. Then the medium was changed to that containing free 450 and 900 nM C_60_, Dox or C_60_-Dox nanocomplexes in 450 nM Dox-equivalent concentration for 24 h, irradiated (10 J/cm^2^ 405 nm LED) as indicated above, incubated for 1 and 3 h and washed once with PBS. Five µM DCFH-DA was added and the fluorescence (λ_ex_ = 488 nm, λ_em_ = 520 nm) was recorded every 5 min over 50 min with the microplate reader Tecan Infinite M200 Pro (Männedorf, Switzerland). At 60 min of incubation, fluorescence images of cells were obtained with the fluorescence microscope Keyence BZ-9000 BIOREVO (Osaka, Japan), equipped with green filter (λ_ex_ = 472 nm, λ_em_ = 520 nm).

### 2.9. Intercellular ATP Content

CCRF-CEM cells were seeded into 96-well plates (10^4^ cells/well) and incubated for 24 h. Cells were treated with 450 and 900 nM free C_60_, Dox or C_60_-Dox nanocomplexes in 450 nM Dox-equivalent concentration for 24 h, irradiated with 405 nm, 10 J/cm^2^ and transferred to 50 µL glucose-free RPMI. At 3 h after light exposure the cell membrane integrity and cellular ATP level were estimated with the Promega Mitochindrial ToxGlo™ assay kit (Madison, WI, USA) according to the manufacturer’s instructions. Briefly, plates were equilibrated to RT for 10 min and to each well the ATP Detection Reagent (50 µL) was added containing luciferin, ATPase inhibitors and thermostable luciferase. After shaking at 600 rpm for 1 min the luminescence intensity was measured with the microplate reader Tecan Infinite M200 Pro (Männedorf, Switzerland).

### 2.10. Caspase 3/7 Activity

CCRF-CEM cells were seeded into 96-well plates (10^4^ cells/well) and incubated for 24 h. Cells were treated with 450 and 900 nM free C_60_, Dox or C_60_-Dox nanocomplexes in 450 nM Dox-equivalent concentration for 24 h and irradiated (405 nm, 10 J/cm^2^) as described above. Activity of caspases 3/7 was determined at 3 h after light exposure using the Promega Caspase-Glo^®^ 3/7 Activity assay kit (Madison, WI, USA) according to the manufacturer’s instructions. Briefly, plates were removed from the incubator and allowed to equilibrate to RT for 30 min. After that, an equal volume of Caspase-Glo 3/7 reagent containing luminogenic peptide substrate was added followed by gentle mixing with a plate shaker at 300 rpm for 1 min. The plate was then incubated at RT for 2 h. The luminescence intensity of the products of caspase 3/7 reaction was measured with the microplate reader Tecan Infinite M200 Pro (Männedorf, Switzerland).

### 2.11. Flow Cytometry Analysis 

CCRF-CEM cells were seeded onto 6-well plates Sarstedt (Nümbrecht, Germany) at a cell density of 2 × 10^5^ cells/well in 2 mL of culture medium, incubated for 24 h, than treated with 450 and 900 nM free C_60_, Dox or C_60_-Dox nanocomplexes in 450 nM Dox-equivalent concentration for 24 h and irradiated (405 nm, 10 J/cm^2^) as described above. At 6 h of incubation period cells were harvested. Apoptosis was detected by Annexin V-FITC/propidium iodide (PI) apoptosis detection kit (eBioscience™, San Diego, CA, USA) according to the manufacturer’s instructions. Briefly, cells were harvested and washed with Binding buffer. After addition of Annexin V-FITC cells were incubated for 15 min at RT in dark. Cells were washed with Binding buffer and at 10 min after PI addition were analyzed (λ_ex_ = 488 nm, λ_em_ (Annexin V-FITC) = 530/40 nm and λ_em_ (PI) = 692/40 nm) with a flow cytometer BD FACSJazz™ (Franklin Lakes, New Jersey, USA). A minimum of 2 × 10^4^ cells per sample were acquired and analyzed with the BD FACS™ software (Franklin Lakes, NJ, USA).

### 2.12. Statistics

All experiments were carried out with a minimum of four replicates. Data analysis was performed using the GraphPad Prism 7 Software (GraphPad Software Inc., San Diego, CA, USA). Paired Student’s t-test was performed. Differences with *p*-values <0.01 were considered to be significant.

The combination index (CI), calculated according to the Chou–Talalay method [51] with the ComboSyn software (ComboSyn, Inc., Paramus, NJ, USA), was used to evaluate pharmacodynamic interactions between non-irradiated C_60_-Dox nanocomplexes and photoexcitation of C_60_ in cells treated with C_60_-Dox nanocomplexes and irradiated with 5 and 10 J/cm^2^ LED light. The following equation was used.
(1)$$CI=\frac{{\left(D\right)}_{1}}{{\left(D_{25}\right)}_{1}}+\frac{{\left(D\right)}_{2}}{{\left(D_{25}\right)}_{2}}$$
where (*D*_25_)_1_ is the concentration of C_60_-Dox that inhibited cell viability to 25%; (*D*_25_)_2_ is the concentration of free C_60_ that inhibited cell viability to 25% after photoexcitation; (*D*)_1_ and (*D*)_2_ are the concentrations of Dox and C_60_ in the C_60_-Dox nanocomplexes which inhibited cell viability to 25% after C_60_ photoexcitation. A CI value of <1, =1 and >1 indicates a synergistic, additive and antagonistic interaction, respectively.

## 3. Results

### 3.1. Localization of C_60_ and Dox in Cells Treated with C_60_-Dox Nanocomplexes

With the use of fluorescence-based techniques we could explain intracellular localization of C_60_ and Dox after CCRF-CEM cells’ treatment with C_60_-Dox nanocomplexes in the 1:1 and 2:1 nanomolar ratio. Since Dox possesses a strong absorption and fluorescence in the visible spectral region [52,53] the direct tracking of this molecule is possible, whereas C_60_ monitoring requires additional immunofluorescence staining [30,31,54].

CCRF-CEM cells were incubated for 24 h with the agents under study and subjected to staining. The overlap of the Dox red signal with the nuclear marker DAPI blue signal confirmed the drug’s nuclear localization (Figure 4). We detected substantially enhanced Dox level in cells treated with C_60_-Dox nanocomplexes as compared with cells treated with the free Dox. Monitoring of the C_60_ distribution by the immunofluorescence green signal confirmed the intracellular accumulation of the nanostructure and pointed to its extranuclear localization. The C_60_ localization within mitochondria accounts for 72% of the whole cellular content as was shown before [31,32]. The observed intracellular allocation of Dox and C_60_ evidenced the effective intracellular Dox release from C_60_-Dox nanocomplexes.

Localization of the nanocomplex components in the different cell compartments strongly supports the possibility of an anticancer double hit strategy that is suggested to be realized by photoinduced pro-oxidant activity of C_60_ in mitochondria [31] and antiproliferative action of Dox in nuclei [2,3,4].

### 3.2. Cell Viability

The viability of cells incubated without any treatment was taken as 100% (control). No effect of C_60_ introduced alone on leukemic cell viability was detected, while the concentration-dependent toxic effect of free Dox was observed. After the treatment with 50, 150, and 450 nM Dox, cell viability was decreased to 81% ± 5%, 70% ± 3%, and 49% ± 5%, correspondingly (Figure 4). When cells were treated in the dark with the C_60_-Dox nanocomplexes at Dox equivalent concentrations, further increase of the Dox toxicity by 10–20% (Figure 5) and the decrease of its IC_50_ (Table 1) were observed. These data denote C_60_’s ability to facilitate intracellular Dox accumulation [55] and, therefore, potentiate its toxic effects.

After combined treatment with C_60_-Dox nanocomplexes and light cells, viability as well as IC_50_ values were considerably decreased compared with their dark toxic effects. The toxicity was dependent on the light fluence and C_60_ concentration in the nanocomplex. Thus, the decrease of cell viability after the treatment with 1:1 C_60_-Dox nanocomplex and 5 J/cm^2^ light was observed only when the nanocomplex was used at 450 nM C_60_ equivalent concentration (Figure 5a). When the light fluence was increased up to 10 J/cm^2^ the pronounced phototoxic effect became evident at all given concentrations of 1:1 C_60_-Dox nanocomplex (Figure 5b) and the IC_50_ values appeared to be three and nine times lower as compared with the IC_50_ for non-irradiated 1:1 C_60_-Dox nanocomplex and for the free Dox, correspondingly (Table 1).

The viability of cells treated with 2:1 C_60_-Dox nanocomplex and irradiated with 5 J/cm^2^ light was decreased substantially in a concentration dependent manner. The most significant toxic effect was observed after the treatment with 2:1 C_60_-Dox nanocomplex and irradiation with 10 J/cm^2^ light, when the IC_50_ values was estimated to be nine and sixteen times lower as compared with the IC_50_ for non-irradiated 2:1 C_60_-Dox nanocomplex and for the free Dox, correspondingly (Table 1). If comparing with the photodynamic effect from C_60_ alone towards CCRF-CEM cells at the same conditions [31], we can conclude that the IC_50_ of photoexcited 2:1 C_60_-Dox was 7 × 10^3^-fold decreased. No signs of appreciable viability were detected when cells were treated with 2:1 C_60_-Dox nanocomplex at 900 nM C_60_ equivalent concentration and irradiated with 10 J/cm^2^ light (Figure 5b).

To estimate the pharmacodynamics interactions of C_60_’s dual functionality, as a drug nanocarrier and as a photosensitizer in cells treated with nanocomplexes, we calculated the value of the combination index (Table 2). When cells were co-treated with C_60_-Dox nanocomplexes and LED light at 5 J/cm^2^ fluence the CI values indicated a synergistic effect. While after co-treatment with 1:1 and 2:1 C_60_-Dox nanocomplexes and LED light at 10 J/cm^2^ fluence, the CI value proved to have a strong and a very strong synergistic effect of the photoexcited nanocomplex components respectively in the applied bimodal strategy of cell treatment.

Next the intracellular ROS generation, ATP level, and the markers of apoptotic death were assessed in CCRF-CEM cells after combined treatment with C_60_-Dox nanocomplexes and light.

### 3.3. Intracellular Reacrive Oxygen Species Generation

The efficient and continuous intracellular ROS production is considered to be a critical step in realization of a photoexcited C_60_ anticancer effect [31,36,40,56,57]. ROS generation in cells was evaluated with the use of the fluorescence dye DCFH-DA [58,59] at 1 and 3 h of incubation after light irradiation or in the dark. The minor increase of the fluorescence signal was detected during the incubation of the control untreated cells (Figure 6). No reliable changes in ROS generation in comparison with the control were observed when 450 or 900 nM C_60_ was applied alone. Treatment with the free 450 nM Dox or C_60_-Dox nanocomplexes was followed by a slight increase of ROS generation at 1 h that was attenuated at 3 h (Figure 6a). When cells treated with the free C_60_ or C_60_-Dox nanocomplexes were irradiated with 405 nm light at 10 J/cm^2^ fluence the ROS production was intensified at both 1 and 3 h. In cells treated with 1:1 or 2:1 C_60_-Dox nanocomplex ROS levels at 3 h after irradiation exceeded the control level by 3.8 times and 5.0 times, correspondingly (Figure 6c).

The analysis of the fluorescence microscopy images (Figure 6b,d) confirmed the obtained quantitative data on intense ROS production in cells irradiated after the treatment with C_60_-Dox nanocomplexes and supports oxidative stress as a precondition of mitochondrial dysfunction and intrinsic apoptotic pathway induction [17,60].

### 3.4. Intracellular ATP Content

Mitochondria play a leading role in apoptosis induction and progression and are an important subcellular target for many photosensitizing drugs [17,60]. Cytotoxic effects of photosensitizers are considered to be realized particularly through the mitochondrial oxidative damage [26,31,37]. Therefore, next we assessed whether the treatment of cells with C_60_-Dox nanocomplexes and light affected ATP production as the main mitochondrial function. Neither free C_60_ and Dox nor light irradiation alone had any effect on the ATP level in cells (Figure 7a).

In cells treated with 1:1 and 2:1 C_60_-Dox nanocomplexes the ATP level was slightly decreased to 84% ± 5% and 87% ± 3% of the control, respectively. The appreciable drop in intracellular ATP level to 30% ± 4% and 28% ± 3% was observed after 10 J/cm^2^ light irradiation of cells treated with 1:1 and 2:1 C_60_-Dox nanocomplexes respectively (Figure 7a), indicating inhibition of mitochondria function that could be attributed to impaired redox balance in cells.

### 3.5. Apoptosis Induction

Apoptotic program execution requires the coordinated activation of multiple subprograms including caspases cascade [5,61]. The executive caspase 3/7 activation and phosphatidylserine translocation into the outer layer of plasma membrane lipid bilayer were evaluated as the markers of apoptotic cell death. No effect of either free C_60_ or Dox as well as of light irradiation alone on caspase 3/7 was observed following 3 h of cells incubation (Figure 7b). Irradiation of cells treated with 450 or 900 nM C_60_ was followed by 1.6-fold and 1.9-fold increase of caspase 3/7 activity, respectively, while after irradiation of cells treated with 1:1 or 2:1 C_60_-Dox nanocomplexes 4.7-fold and 5.8-fold increase of caspase 3/7 activity respectively compared with control was observed (Figure 7b).

Finally, we checked the exposure of phosphatidylserine on the cell surface as an “eat me” signal, that induces phagocytic recognition and destruction of apoptotic cells [62]. To differentiate apoptotic cells fluorescence activated cell sorting (FACS) was used. On FACS histograms (Figure 8) the cell distribution at 6 h after the treatment with either free C_60_ or Dox or C_60_-Dox nanocomplexes is presented according to the green and red fluorescence intensities of Annexin V-FITCI and PI respectively. Viable (Annexin V-FITC negative, PI negative), early apoptotic (Annexin V-FITC positive, PI negative), late apoptotic (Annexin V-FITC positive, PI positive), and necrotic (Annexin V-FITC negative, PI positive) cells in their quantitative populations analyses are presented in Figure 8b. Neither treatment with C_60_ nor 405 nm light irradiation alone had a significant effect on cell distribution profiles (FACS histograms are not shown). A slight increase in the number of early apoptotic cells was observed after treatments with free Dox, C_60_-Dox nanocomplexes or photoexcited C_60_. When cells treated with C_60_-Dox nanocomplexes were exposed to light a distribution-shift towards late apoptosis was observed. Thus, the content of Annexin V-FITC positive and PI positive cells in population of cells treated with photoexcited 1:1 and 2:1 C_60_-Dox nanocomplexes reached 93% and 96%, correspondingly (Figure 8).

Taken together the data obtained confirmed the pro-apoptotic effect of combined treatment with C_60_-Dox nanocomplexes and 405 nm LED light irradiation on leukemic cells.

## 4. Discussion

The nanoparticulation of anticancer drugs expands the scope of their chemical behavior and pharmacodynamics, as well as reducing efficient doses and unwanted side effects. Inclusion of the anticancer drug Dox into nanosized delivery systems prolongs its retention in the organism and favors its targeted accumulation in cancer cells [63,64].

In previous studies we exploited the ability of C_60_ nanostructure’s polyaromatic surface to absorb aromatic Dox molecules and synthesized non-covalent C_60_-Dox nanocomplexes. The physicochemical properties of the C_60_-Dox nanocomplexes studied with the different analytical methods confirmed their stability and biological applicability [45,52,59,65]. Complexation with C_60_ increases the intracellular Dox level and improves Dox efficiency against human leukemic and colon cancer cells [49,55].

In the current study combination of the chemotherapeutic and photodynamic treatment strategies was explored on the basis of Dox nanocomplexes with C_60_. Nanocomplex of C_60_ and Dox at molar ratios 1:1 and 2:1 and at Dox equivalent concentrations in a range ≤IC_50_ (150–450 nM) were tested on human leukemic CCRF-CEM cells in combination with light irradiation.

With the use of indirect C_60_ immunostaining we confirmed the intracellular accumulation of the carbon nanostructure in cells treated with free C_60_ or C_60_-Dox nanocomplexes. When cells were treated with C_60_-Dox nanocomplexes, C_60_ was found to be localized in the extranuclear space assumed to be in mitochondrial membranes as it was shown for C_60_ at higher concentrations [31,32], whereas Dox was accumulated in the cell nucleus and in a higher concentration than after treatment with a free drug. These data are linked to the C_60_ ability to promote passive diffusion and/or endocytosis/pinocytosis of the small molecules in cancer cells [66,67,68] and to bind P-glycoproteins [69], inhibiting Dox’s pumping out from the cell. Comparably, graphene-based triple delivery nanosystems non-covalently loaded with Dox and phthalocyanine ensured higher cellular drug uptake and effective intracellular drug release [21].

The allocation of C_60_ and Dox inside leukemic cells sets a background for the application of the “two swords” treatment strategy based on CT and PDT combination. Thus, nuclear Dox intercalated into DNA is supposed to block its transcription, replication, and repair [2,3,4], whereas photoexcited extranuclear C_60_ can produce ROS and induce mitochondrial pathway of apoptosis [26,31,36,37,70].

The C_60_ absorption spectrum has three intense bands in the ultraviolet region and a long broad tail up to the red region of the visible light [31,43]. The mutagenic potential of ultraviolet light makes its application unfavorable, therefore, we used irradiation with a 405 nm high power single chip light emitting diode at the fluence ≤10 J/cm^2^, that was shown previously to be nontoxic and efficient for the photoexcitation of C_60_ accumulated in CCRF-CEM cells [31].

In order to exploit C_60_ photosensitizing activity a high-power a single chip light-emitting diode-based light source was constructed. The use of high-power single chip LEDs is expected to promote PDT application, since they have a higher portability and an extremely lower cost, compromising the efficiency of lasers [71,72]. The possibility to vary the evenly irradiated area is one of the main advantages of the developed system. It provided the same power density at any irradiation point, allowing for accurate calculation and selection of the irradiation dose. Consequently, the optical elements greatly increased the efficiency of using LED irradiation and helped to collimate irradiation in a narrower beam. To address a challenge of limited penetration depth of blue light in biological tissue, next steps could be aimed at additional skin optical clearing [73,74] or coupling of the LED light source with fiber optics for direct and precise light intra-tissue delivery.

It should be noted that we used C_60_ in nanomolar concentrations in contrast to micromolar application in our previous studies [26,31,36,37] and by other authors [38,56,57,68,75,76]. Nanomolar C_60_ was shown to have no dark toxicity and a slight pro-oxidant effect with 11.5% decrease of leukemic cell viability after 405 nm LED light irradiation at 10 J/cm^2^ fluence.

When leukemic cells were treated with a 2:1 C_60_-Dox nanocomplex and irradiated with 10 J/cm^2^ 405 nm LED light a 16-fold decrease of IC_50_ was observed as compared with the IC_50_ value for the free Dox (390 nM). Phototoxicity of 1:1 C_60_-Dox nanocomplex occurred to be less pronounced causing a 9-fold decrease of IC_50_ that can be attributed to the lower content of C_60_ as a photosensitizer. The value of the combination index, which was calculated to characterize the pharmacodynamic interactions, indicated a strong synergy between non-irradiated 2:1 C_60_-complexed Dox and photoexcited C_60_ with 10 J/cm^2^ light. The high pro-apoptotic efficiency of the C_60_-Dox nanocomplexes and light irradiation against CCRF-CEM cells were confirmed by significant increase of intracellular ROS, decreased ATP levels, caspase 3/7 activation, and transition of 90% of cells to the late apoptosis stage.

The data obtained demonstrate effective combination of chemotherapeutic and photodynamic cancer treatment strategies on the basis of the C_60_-Dox nanosystem. To our knowledge we were the first to apply nanomolar concentrations of photosensitizer and drug in vitro for toxicity optimization in the frame of photodynamic chemotherapy with carbon nanomaterial. According to the recent literature data on carbon nanoparticle-mediated photodynamic chemotherapy the synergistic toxic effect of drug-loaded graphene was achieved with light irradiation of cervix, breast, and skin cancer cells treated with Dox-polylysine graphene–phthalocyanine [21] or lung cancer cells treated camptothecin–graphene oxide–hypocrellin A [77], both in the µM Dox concentrations. Moreover, dual functionality of C_60_, as a drug nanocarrier and as a photosensitizer, enabled an easy and fast preparation of two-component C_60_-Dox as compared to the mentioned three-component graphene-based nanosystems.

Chemo- and photodynamic anticancer agents have distinct intracellular targets and, therefore, induce different signaling pathways of cell injury. Earlier studies showed tumor-specific differential effects of agents under study. Thus, Dox attacked specifically fast proliferating cells [2,3,4], whereas C_60_ mainly targeted the redox state of the cell [24,25,28]. The synergistic effect of PDT and CT combination is attributed mainly to the further amplification of oxidative stress [8]. Intensive ROS production promotes apoptosis and assists drug delivery due to ROS-mediated lipid peroxidation of endosome membranes [8,16,17]. The efflux of the drug can be inhibited as well by ROS-mediated oxidation of the intracellular domain of the multidrug resistance P-glycoprotein [78,79,80]. ROS as signal intracellular messengers shift profiles of signaling pathways in treated cells. Hence, carbon nanoparticle induced ROS-mediated activation of the mitogen-activated protein kinase, that increased the vulnerability of lung cancer cells towards paclitaxel [81]. The cooperative enhancement interactions between mechanisms of chemo- and photodynamic therapies contribute to the obtained synergistic effect (namely “1 + 1 >2”).

## Figures and Tables

**Figure 1 nanomaterials-09-01540-f001:**
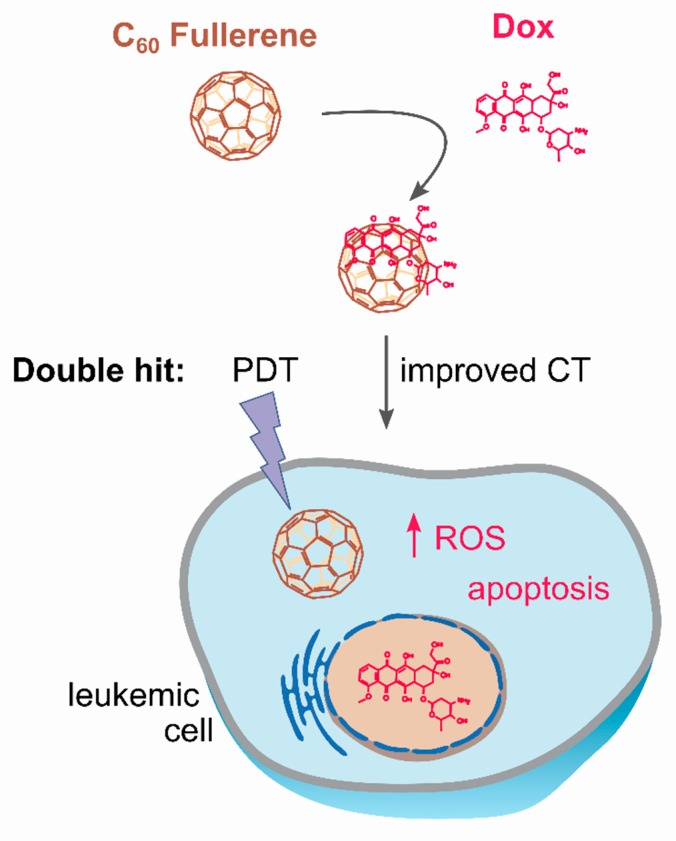
C_60_-Dox nanocomplex for photodynamic chemotherapy: C_60_ delivers Doxorubicin (Dox) into leukemic cells and intensifies its accumulation; following internalization, the drug is anticipated to be released from the nanocomplex; cancer cell is exposed to the double cytotoxic hit from both photoexcited C_60_ (photodynamic therapy, PDT) and co-delivered Dox (improved chemotherapy, CT).

**Figure 2 nanomaterials-09-01540-f002:**
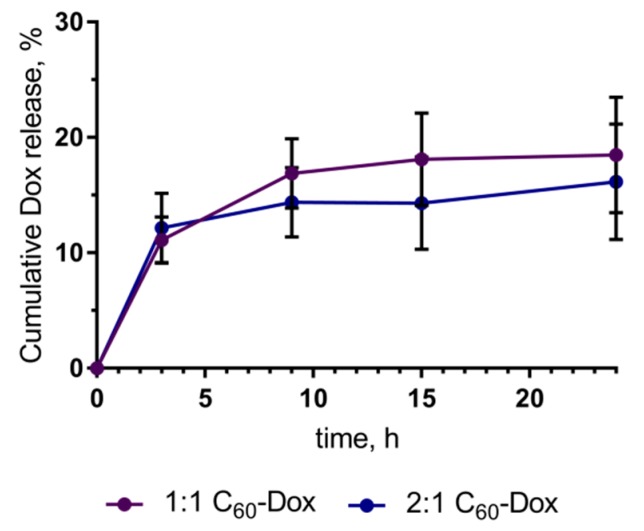
Dox release from C_60_-Dox nanocomplexes during 24 h of incubation in Roswell Park Memorial Institute (RPMI 1640) medium.

**Figure 3 nanomaterials-09-01540-f003:**
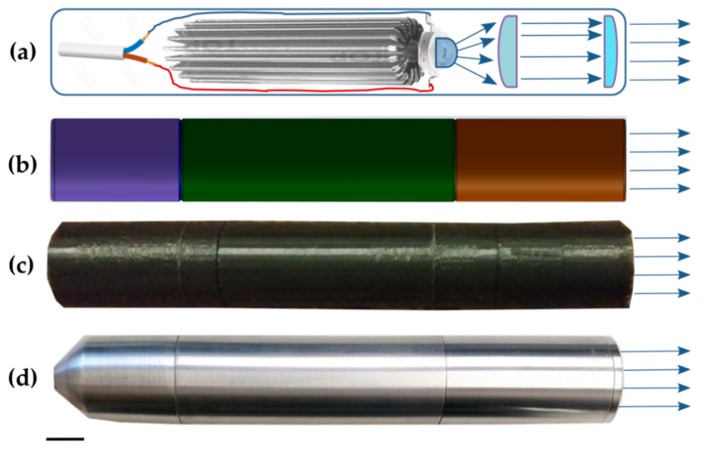
LED light system: (**a**) scheme, that reveals its electrical part, LED, and optical system, (**b**) design of the mounting carcass in 3D Software SOLIDWorks (Dassault Systems, Vélizy-Villacoublay, France), (**c**) 3D printed plastic 1st model, (**d**) final metal model; scale bar corresponds to 10 mm.

**Figure 4 nanomaterials-09-01540-f004:**
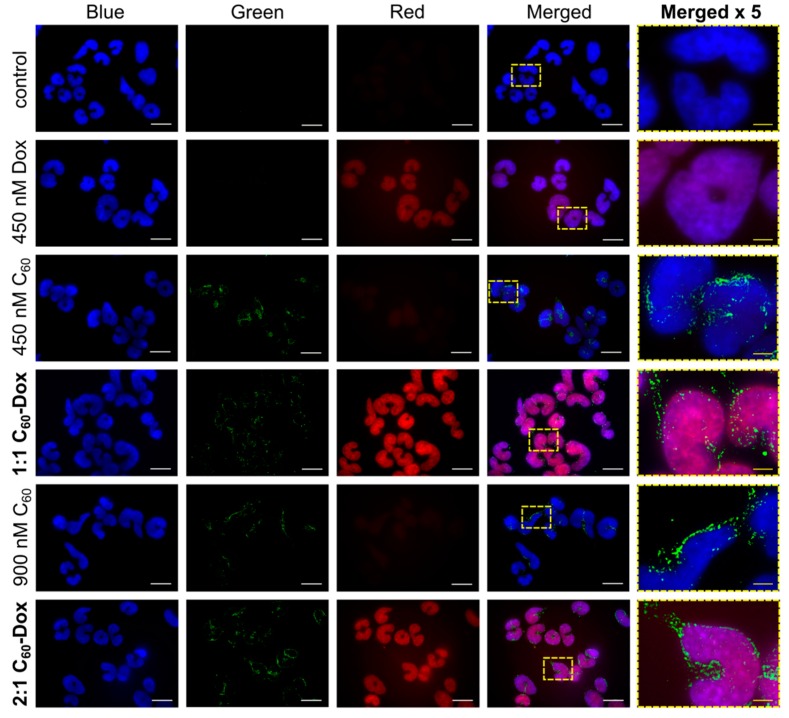
Fluorescence microscopy images of CCRF-CEM cells, stained with 4′,6-diamidine-2′-phenylindole dihydrochloride (DAPI, Blue), fluorescein isothiocyanate-based immunostaining for C_60_ (Green) and Dox (Red) after treatment with: 450 and 900 nM C_60_, 450 nM Dox, 1:1 or 2:1 C_60_-Dox nanocomplex. The white scale bar corresponds to 20 µm; the yellow scale bar on images in the column “Merged x5” corresponds to 4 µm.

**Figure 5 nanomaterials-09-01540-f005:**
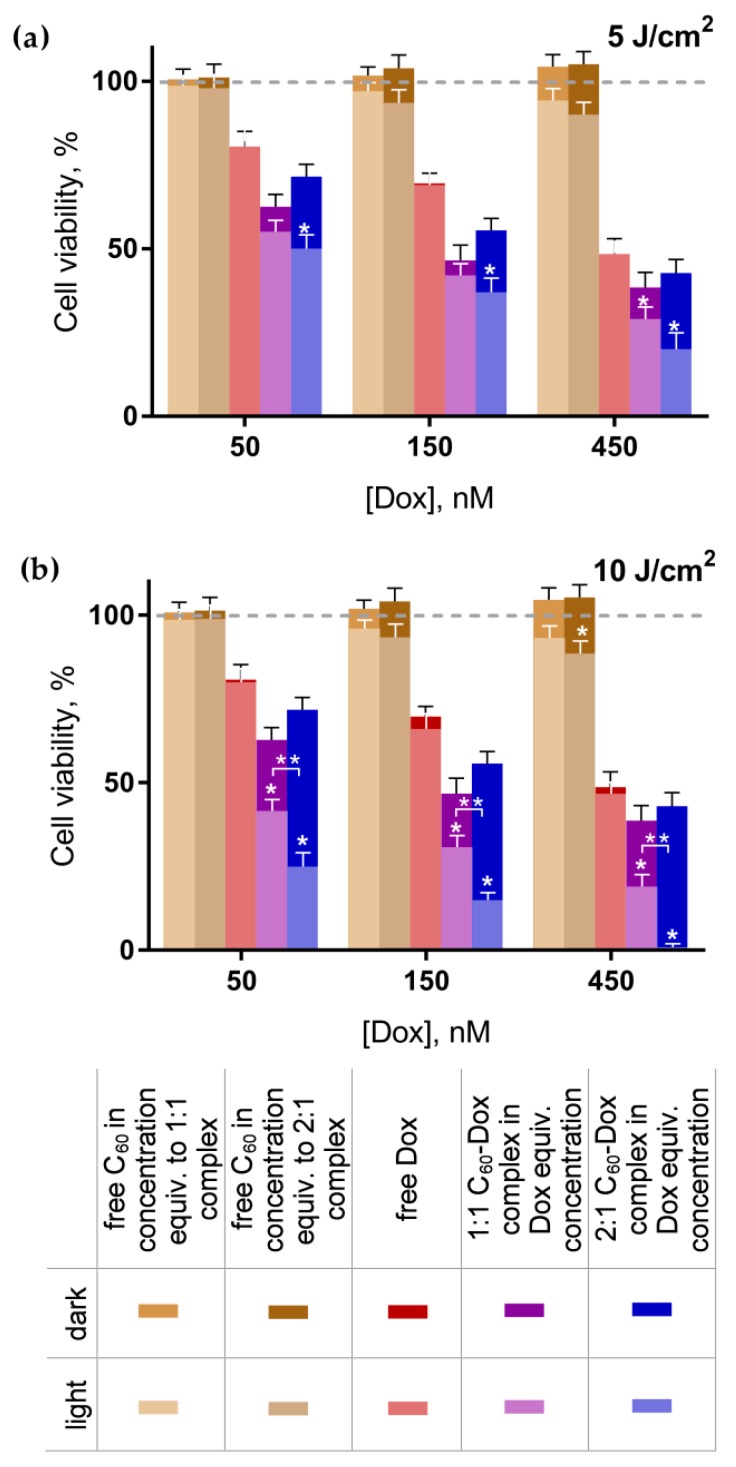
Viability of CCRF-CEM cells: cells were treated with either C_60_ and Dox alone or C_60_-Dox nanocomplexes in Dox equivalent concentrations and incubated in the dark or after light irradiation with 405 nm LED at 5 J/cm^2^ (**a**) or 10 J/cm^2^ (**b**) (* *p* ≤ 0.01 in comparison with the respective dark control, ** *p* ≤ 0.01 in comparison with the photoexcited 1:1 nanocomplex).

**Figure 6 nanomaterials-09-01540-f006:**
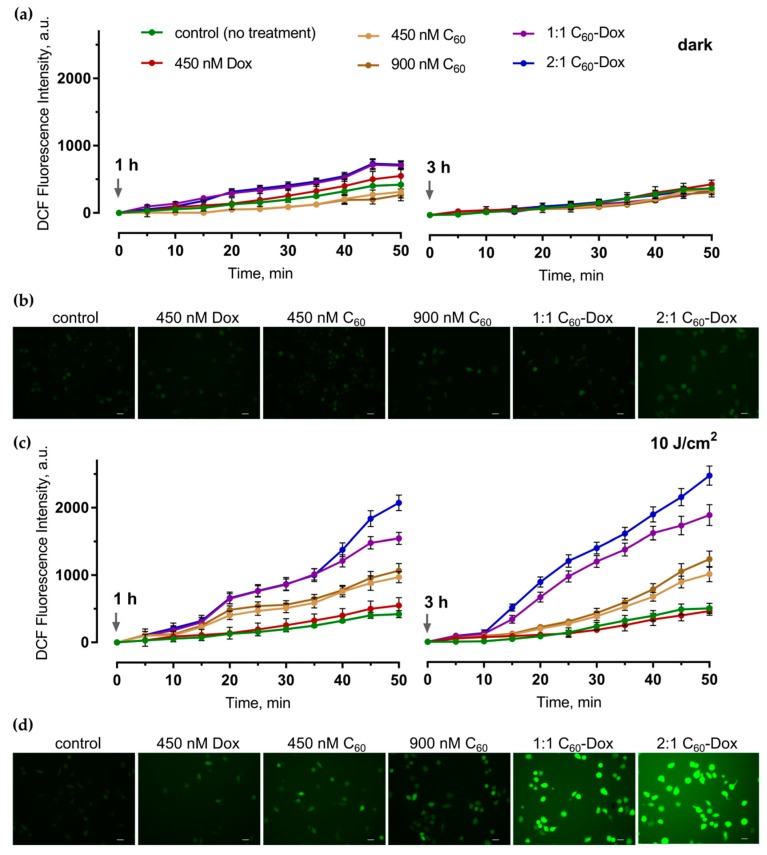
Reacrive oxygen species generation in CCRF-CEM cells treated with either Dox or C_60_ alone or with C_60_-Dox nanocomplexes: the dynamics of ROS generation in cells at 1 and 3 h after the treatment in the dark (**a**) or irradiation with 10 J/cm^2^ 405 nm LED (**c**); the fluorescence microscopy images of cells at 3 h after the treatment in dark (**b**) or light irradiation (**d**) and further 60 min incubation with 2,7-dichlorofluorescin diacetate (DCFH-DA); scale bar corresponds to 20 µm.

**Figure 7 nanomaterials-09-01540-f007:**
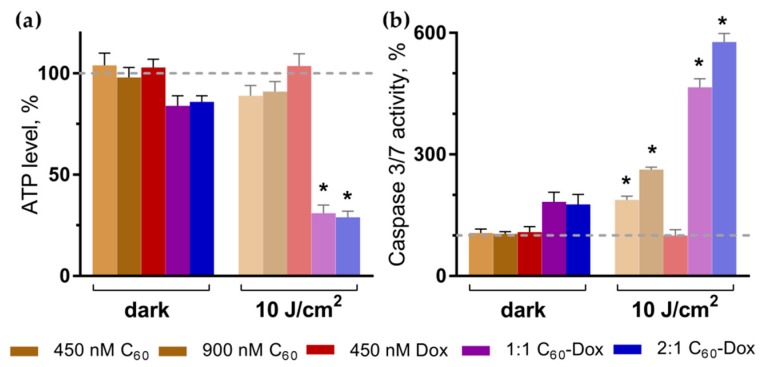
ATP level (**a**) and caspase 3/7 activity (**b**) in CCRF-CEM cells at 3 h after treatment. Treatment was done with either free C_60_ and Dox or C_60_-Dox nanocomplexes in the dark or after irradiation with 10 J/cm^2^ 405 nm LED (* *p* ≤ 0.01 in comparison with the respective dark control).

**Figure 8 nanomaterials-09-01540-f008:**
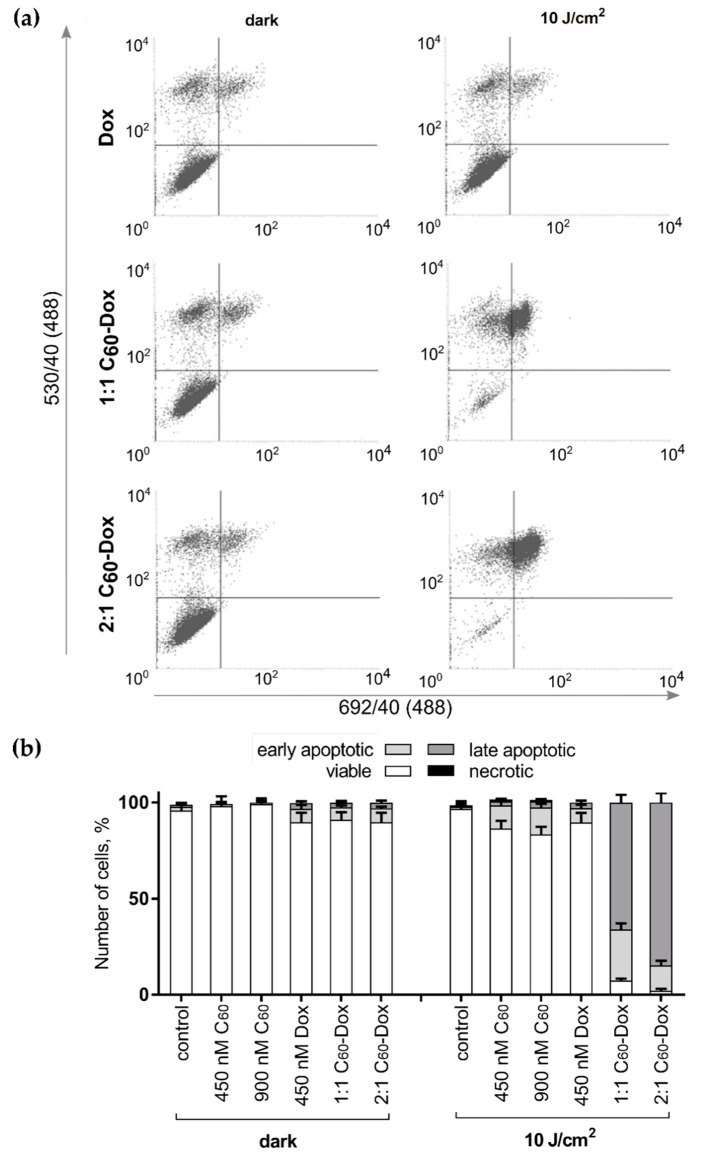
Cell death differentiation in CCRF-CEM treated with either free C_60_, Dox or C_60_-Dox nanocomplexes: (**a**) flow cytometry histograms of CCRF-CEM cells stained with Annexin V-FITC/ propidium iodide (PI) after treatment either with C_60_-Dox alone or in combination with 405 nm light (in each panel the lower left quadrant shows the content of viable, upper left quadrant—early apoptotic, upper right quadrant—late apoptotic, lower right quadrant—necrotic cells populations); (**b**) Quantitative analysis of cell population content, differentiated with double Annexin V-FITC/PI staining.

**Table 1 nanomaterials-09-01540-t001:** IC_50_ values for the free Dox and C_60_-Dox nanocomplexes.

IC_50_, nM	Dark	5 J/cm^2^	10 J/cm^2^
Dox	390 ± 56	382 ± 53	336 ± 49
1:1 C_60_-Dox	135 ± 29	86 ± 19	44 ± 7 *
2:1 C_60_-Dox	225 ± 34 **	64 ± 11 *	25 ± 4 *^,^**

* *p* ≤ 0.01 in comparison with the respective dark control, ** *p* ≤ 0.01 in comparison with the 1:1 nanocomplex.

**Table 2 nanomaterials-09-01540-t002:** Combination index of interaction between phototoxic effects of C_60_ (PDT) and non-irradiated C_60_-Dox nanocomplexes (improved CT) ^1^.

CI	5 J/cm^2^	10 J/cm^2^
1:1 C_60_-Dox	0.546 ± 0.037 (synergism)	0.130 ± 0.009 (strong synergism)
2:1 C_60_-Dox	0.316 ± 0.023 (synergism)	0.097 ± 0.002 (very strong synergism)

^1^ This was measured after cells’ co-treatment with C_60_-Dox nanocomplex and LED light. Classification of interaction was determined according to Chou [51].

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
