# Peer review of "Synergy of Chemo- and Photodynamic Therapies with C_60_ Fullerene-Doxorubicin Nanocomplex"

_nanomaterials, 2019, doi:10.3390/nano9111540_

Round 1
Reviewer 1 Report
I write you in regards to manuscript entitled “Synergy of Chemo- and Photodynamic Therapies with C60 Fullerene-Doxorubicin Nanocomplex” which you submitted to Nanomaterials.
As author notes in this report, this study might be useful information for the mechanism of Chemo- and Photodynamic Therapies. This is a well-written and useful contribution, which I think is entirely suitable for publication after minor revision.
・Figure 7 (b) is difficult to understand. In particular, bar graph in the dark group is deep in color.
・Chemo- and Photodynamic Therapies with C60 was effective for CCRF-CEM cells in vitro. However, C60 should be excited by 405 nm light, which is absorbed by hemoglobin. I think it is difficult to treat leukemic cells in blood with 405 nm LED light. Therefore, in introduction, you should explain the reason why you use leukemic cells for this study.
Author Response
Dear Reviewer 1,
On behalf of all co-authors, I am thanking you for the constructive revision of our manuscript entitled “Synergy of Chemo- and Photodynamic Therapies with C60 Fullerene-Doxorubicin Nanocomplex” (Manuscript ID: nanomaterials-618446).
The authors are grateful to you for the thoughtful comments and suggestions, that help to enhance the quality of this manuscript. Based on the received comments, we have carefully revised our manuscript text as well as Figure 7 (b). All new changes were highlighted in red in the manuscript. A point-by-point response with further action from our side in a response to the review is included below.
Response to Reviewer 1 Comments
Point 1: Figure 7 (b) is difficult to understand. In particular, bar graph in the dark group is deep in color.
Response 1: Thank you for your advice! This note was taken into account and the graph presented on Figure 7 (b) was adjusted respectively.
Point 2: Chemo- and Photodynamic Therapies with C60 was effective for CCRF-CEM cells in vitro. However, C60 should be excited by 405 nm light, which is absorbed by hemoglobin. I think it is difficult to treat leukemic cells in blood with 405 nm LED light. Therefore, in introduction, you should explain the reason why you use leukemic cells for this study.
Response 2: Thank you for your meaningful recommendation! We addressed the concern of the Reviewer with provided explanation below and addition of the information in the abstract.
Leukemic cell lines are common models of human cancer for experimental investigations on the cellular level. Leukemia, cancer of the body's blood-forming tissues, including the bone marrow and the lymphatic system, reached 437 033 new diagnosed cases in 2018 [Bray, F.; Ferlay, J.; Soerjomataram, I.; Siegel, R.L.; Torre, L.A.; Jemal A. Global cancer statistics 2018: GLOBOCAN estimates of incidence and mortality worldwide for 36 cancers in 185 countries. CA Cancer J Clin 2018, 68(6), 394-424. doi:10.3322/caac.21492], that accounted in average 14 out of 100,000 persons per year. Thus, the human leukemic CCRF-CEM cells were chosen as a main in vitro cancer model for the current research. Since, the presented manuscript delivers only in vitro results, where culturing conditions include no hemoglobin, CCRF-CEM cell line could be considered as a reliable experimental model to assess treatment mechanism in vitro. As CCRF-CEM cells were the first cell line being tested, now we are extending our study to check how efficient is the proposed treatment strategy on cancer cells with different origin. Our further research plan includes as well transfer on in vivo animal models, where leukemia would not be 1st choice indeed. For animal models, we agree with the Reviewer’s comment not only because of the hemoglobin high absorption of the used light but also because the main leukemic blood-forming tissue’s localization in the bone marrow, that is highly challengeable to be exposed to any light irradiation. We thank the Reviewer for raising important point, that would be addressed during our further study on animal model where hemoglobin is presented.
Thank you for your time!
Sincerely yours, Anna Grebinyk

Reviewer 2 Report
This manuscript describes the preparation and anticancer properties of nano-sized particles composed of C60 with adsorbed doxorubicin (Dox). The investigators show that the particles have two mechanisms for killing cells. The C60-Dox particles passively release free Dox inside the cell and, when exposed to 405 nm light, they produce ROS that kills the cell. The manuscript is generally well written and it has explanations and descriptions that are easy to follow. The authors and editor should consider the following comments and suggestions prior to acceptance and publication.
The wavelength of radiation is in the blue region (405 nm) of the spectrum which is not optimal for tissue penetration. The authors should comment on this fact and say how this may limit the application of the particles in chemotherapy. There are no errors for the data given in Table 2, “Combination index of interaction between phototoxic effects of C60 (PDT) and non-irradiated C60-Dox nanocomplexes (improved CT).” Without errors, one cannot tell if the values are different from one another. What is the surface density of Dox on the C60 particle in the two formulations, i.e., how much of the surface is covered by Dox? It would seem that this could be estimated from the surface area of C60 and the size of Dox. The surface density is not stated in the manuscript. What is the stability of the C60-Dox particle, i.e., does Dox diffuse away from C60 when C60-Dox is suspended in water, normal saline or culture medium? Since the Dox molecules are adsorbed on the surface through Pi-Pi interactions, drug binding to C60 is is an equilibrium process. If the ultimate goal is to develop the C60-Dox particles into a practical drug, this information would be important to know. I read the earlier paper by this research group (Frohme, and coworkers, Complexation with C60 Fullerene Increases Doxorubicin Efficiency against 638 Leukemic Cells In Vitro. Nanoscale Res Lett 2019, 14, 61). In the article the researchers say, “The stability (ζ-potential value) and size distribution (hydrodynamic diameter) [20, 36–39] of complexes were systematically checked and shown to be practically unchanged after 6 months of storage in physiological saline solution.” However, they did not address the issue of drug diffusing away from C60. If a drug formulation is envisioned, one would like to know the stability of the C60-Dox particles which is directly related to the binding constant of Dox toward C60. This issue needs to be directly addressed in the manuscript.
Author Response
Dear Reviewer 2,
On behalf of all co-authors, I am thanking you for the constructive revision of our manuscript entitled “Synergy of Chemo- and Photodynamic Therapies with C60 Fullerene-Doxorubicin Nanocomplex” (Manuscript ID: nanomaterials-618446).
The authors are grateful to you for the thoughtful comments and suggestions, that help to enhance the quality of this manuscript. Based on the received comments, we have carefully revised our discussion and extended nanocomplexes characterization. All new changes were highlighted in red in the manuscript. A point-by-point response with further action from our side in a response to the review is included below.
Response to Reviewer 2 Comments
Point 1: The wavelength of radiation is in the blue region (405 nm) of the spectrum which is not optimal for tissue penetration. The authors should comment on this fact and say how this may limit the application of the particles in chemotherapy.
Response 1: Thank you for the meaningful point! We have extended discussion part on the used light source as well as possible improvement options in the text of the manuscript.
We further critically assess C60 PDT here to fully address the matter since C60 alone was not in the focus of the presented study.
The presented study with the use of 405 nm LED as a light source for C60 excitation supports the common trend in recent years to investigate C60 as a potential photosensitizer to mediate PDT of diverse diseases. Light sources usually provided UV, blue, green or white because the C60 absorption is biased towards lower wavelengths with three intense bands in UV region and a broad tail up to the red light [43]. Since in vivo PDT commonly uses red light for its improved tissue penetrating properties, it was unclear whether C60 would mediate effective PDT in vivo. However, such concerns were addressed in a study of intraperitoneal photodynamic C60 therapy on a mouse model of abdominal dissemination of colon adenocarcinoma. Thus, Mroz et al. detected, that mice suffered toxicity after C60 PDT with red light but exhibited a beneficial therapeutic effect after white light illumination suggests that this supposed drawback may actually be an advantage instead [Mroz, P.; Xia, Y; Asanuma, D.; et al. Intraperitoneal photodynamic therapy mediated by a fullerene in a mouse model of abdominal dissemination of colon adenocarcinoma. Nanomed 2011, 7(6), 965-74 doi:10.1016/j.nano.2011.04.007].
Point 2: There are no errors for the data given in Table 2, “Combination index of interaction between phototoxic effects of C60 (PDT) and non-irradiated C60-Dox nanocomplexes (improved CT).” Without errors, one cannot tell if the values are different from one another.
Response 2: Thank you for the correction! We added the missing information to Table 2 respectively.
Point 3: What is the surface density of Dox on the C60 particle in the two formulations, i.e., how much of the surface is covered by Dox? It would seem that this could be estimated from the surface area of C60 and the size of Dox. The surface density is not stated in the manuscript.
Response 3: Thank you for the point! We have extended methodological part adding characterization of the nanocomplexes in the “Methods: C60 and C60-Dox nanocomplex synthesis” part of the manuscript.
The maximum number of ligand molecules (N) that can be bound with C60 fullerene in aqueous solution may be estimated independently with the help of proposed general up-scaled model [47] as well as molecular modeling. The following value was obtained for Dox: N=3 [48,49]. The calculated equilibrium hetero-complexation constant was equal to KL≈6000 M-1 [48].
Point 4: What is the stability of the C60-Dox particle, i.e., does Dox diffuse away from C60 when C60-Dox is suspended in water, normal saline or culture medium? Since the Dox molecules are adsorbed on the surface through Pi-Pi interactions, drug binding to C60 is is an equilibrium process. If the ultimate goal is to develop the C60-Dox particles into a practical drug, this information would be important to know. I read the earlier paper by this research group (Frohme, and coworkers, Complexation with C60 Fullerene Increases Doxorubicin Efficiency against 638 Leukemic Cells In Vitro. Nanoscale Res Lett 2019, 14, 61). In the article the researchers say, “The stability (ζ-potential value) and size distribution (hydrodynamic diameter) [20, 36–39] of complexes were systematically checked and shown to be practically unchanged after 6 months of storage in physiological saline solution.” However, they did not address the issue of drug diffusing away from C60. If a drug formulation is envisioned, one would like to know the stability of the C60-Dox particles which is directly related to the binding constant of Dox toward C60. This issue needs to be directly addressed in the manuscript.
Response 4: Thank you very much for raising important points!
We do agree with the Reviewer that stability is an essential requirement to any nanoformulation. Therefore, the developed nanocomplexes were rigorously studied with various chemical and physical techniques before moving to the in vitro study with biological object.
In a pioneering attempt [48] showed a simple and fast method of C60 noncovalent complexation with Dox in water and later in physiological solution [45]. Molecular modeling, uv-vis spectroscopy, atomic-force microscopy, dynamic light and small-angle X-ray scattering [45-48], high-performance liquid chromatography–mass spectrometry evidenced C60-Dox nanocomplex formation and its stability with the energy –6.3 kcal/mol [48].
Measuring value of the translational diffusion coefficient as a function of C60 concentration at constant Dox concentration was performed in [46]. The diffusion curve displays very distinct changes at small C60 concentrations and reaches a plateau for higher concentrations. It allows to conclude that the binding of Dox molecules mainly occurs with C60 clusters in aqueous solution. Indirectly this conclusion is confirmed by the atomic force microscopy data [46]: most likely C60-Dox nanocomplexes stabilized by the Dox-induced attenuation of electrostatic repulsion between C60.
We would like to note that the research presented in the current paper aims to clarify biological effects of the proposed treatment strategy. Therefore, we would like to address the concern of the Reviewer with the description of our experimental work-flow and respective experimental data (not included in the manuscript, partially published before), that should hopefully weaken the concern.
Dynamic Light Scattering (DLS), uv-vis spectroscopy and high-performance liquid chromatography–mass spectrometry measurements were done for each newly synthesized complex as well as during its storage with monthly repetition. The stability (ζ-potential value) and size distribution (hydrodynamic diameter) of nanocomplexes as described in [45, 55] was systematically checked and shown to be practically unchanged after 6 months of storage in physiological saline solution.
The biological applicability was proven in our recent study where C60-Dox 1:1 and 2:1 nanocomplexes were shown to remain its average hydrodynamic diameter around 135 ± 14 nm and 134 ± 16 nm respectively in physiological saline solution (0.9 % NaCl) up to 6 months [55].
To estimate the stability in cell culture medium 1 µM C60-Dox nanocomplexes were incubated at 370C for 72 h in RPMI supplemented with 10 % FBS (standard cell culture media for CCRF-CEM cell line). The detected stability of the maximum (≤ 140 nm) indicated that there was no additional aggregation of the C60-Dox nanocomplexes and no diffusion of Doc from the nanocomplexes during a prolonged incubation in FBS-supplemented cell culture medium which confirmed their suitability for in vitro study [55].
We believe that the manuscript would not gain substantial new information from the additional experimental data that more specifically aim on storage stability of complexes in physiological saline solution or cell culture media, moreover those data have been already published before. We added the respective references and the detailed description of the stability, binding and diffusion of nanocomplexes in the “Methods: C60 and C60-Dox nanocomplex synthesis” part.
Thank you for your time!
Dear Reviewer 2, On behalf of all co-authors, I am thanking you for the constructive revision of our manuscript entitled “Synergy of Chemo- and Photodynamic Therapies with C60 Fullerene-Doxorubicin Nanocomplex” (Manuscript ID: nanomaterials-618446). The authors are grateful to you for the thoughtful comments and suggestions, that help to enhance the quality of this manuscript. Based on the received comments, we have carefully revised our discussion and extended nanocomplexes characterization. All new changes were highlighted in red in the manuscript. A point-by-point response with further action from our side in a response to the review is included below.
Response to Reviewer 2 Comments
Point 1: The wavelength of radiation is in the blue region (405 nm) of the spectrum which is not optimal for tissue penetration. The authors should comment on this fact and say how this may limit the application of the particles in chemotherapy.
Response 1: Thank you for the meaningful point! We have extended discussion part on the used light source as well as possible improvement options in the text of the manuscript. We further critically assess C60 PDT here to fully address the matter since C60 alone was not in the focus of the presented study. The presented study with the use of 405 nm LED as a light source for C60 excitation supports the common trend in recent years to investigate C60 as a potential photosensitizer to mediate PDT of diverse diseases. Light sources usually provided UV, blue, green or white because the C60 absorption is biased towards lower wavelengths with three intense bands in UV region and a broad tail up to the red light [43]. Since in vivo PDT commonly uses red light for its improved tissue penetrating properties, it was unclear whether C60 would mediate effective PDT in vivo. However, such concerns were addressed in a study of intraperitoneal photodynamic C60 therapy on a mouse model of abdominal dissemination of colon adenocarcinoma. Thus, Mroz et al. detected, that mice suffered toxicity after C60 PDT with red light but exhibited a beneficial therapeutic effect after white light illumination suggests that this supposed drawback may actually be an advantage instead [Mroz, P.; Xia, Y; Asanuma, D.; et al. Intraperitoneal photodynamic therapy mediated by a fullerene in a mouse model of abdominal dissemination of colon adenocarcinoma. Nanomed 2011, 7(6), 965-74 doi:10.1016/j.nano.2011.04.007].
Point 2: There are no errors for the data given in Table 2, “Combination index of interaction between phototoxic effects of C60 (PDT) and non-irradiated C60-Dox nanocomplexes (improved CT).” Without errors, one cannot tell if the values are different from one another.
Response 2: Thank you for the correction! We added the missing information to Table 2 respectively.
Point 3: What is the surface density of Dox on the C60 particle in the two formulations, i.e., how much of the surface is covered by Dox? It would seem that this could be estimated from the surface area of C60 and the size of Dox. The surface density is not stated in the manuscript.
Response 3: Thank you for the point! We have extended methodological part adding characterization of the nanocomplexes in the “Methods: C60 and C60-Dox nanocomplex synthesis” part of the manuscript. The maximum number of ligand molecules (N) that can be bound with C60 fullerene in aqueous solution may be estimated independently with the help of proposed general up-scaled model [47] as well as molecular modeling. The following value was obtained for Dox: N=3 [48,49]. The calculated equilibrium hetero-complexation constant was equal to KL≈6000 M-1 [48].
Point 4: What is the stability of the C60-Dox particle, i.e., does Dox diffuse away from C60 when C60-Dox is suspended in water, normal saline or culture medium? Since the Dox molecules are adsorbed on the surface through Pi-Pi interactions, drug binding to C60 is is an equilibrium process. If the ultimate goal is to develop the C60-Dox particles into a practical drug, this information would be important to know. I read the earlier paper by this research group (Frohme, and coworkers, Complexation with C60 Fullerene Increases Doxorubicin Efficiency against 638 Leukemic Cells In Vitro. Nanoscale Res Lett 2019, 14, 61). In the article the researchers say, “The stability (ζ-potential value) and size distribution (hydrodynamic diameter) [20, 36–39] of complexes were systematically checked and shown to be practically unchanged after 6 months of storage in physiological saline solution.” However, they did not address the issue of drug diffusing away from C60. If a drug formulation is envisioned, one would like to know the stability of the C60-Dox particles which is directly related to the binding constant of Dox toward C60. This issue needs to be directly addressed in the manuscript.
Response 4: Thank you very much for raising important points! We do agree with the Reviewer that stability is an essential requirement to any nanoformulation. Therefore, the developed nanocomplexes were rigorously studied with various chemical and physical techniques before moving to the in vitro study with biological object. In a pioneering attempt [48] showed a simple and fast method of C60 noncovalent complexation with Dox in water and later in physiological solution [45]. Molecular modeling, uv-vis spectroscopy, atomic-force microscopy, dynamic light and small-angle X-ray scattering [45-48], high-performance liquid chromatography–mass spectrometry evidenced C60-Dox nanocomplex formation and its stability with the energy –6.3 kcal/mol [48]. Measuring value of the translational diffusion coefficient as a function of C60 fullerene concentration at constant Dox concentration was performed in [46]. The diffusion curve displays very distinct changes at small C60 fullerene concentrations and reaches a plateau for higher concentrations. It allows to conclude that the binding of Dox molecules mainly occurs with C60 fullerene clusters in aqueous solution. Indirectly this conclusion is confirmed by the atomic force microscopy data [46]: most likely C60-Dox nanocomplexes stabilized by the Dox-induced attenuation of electrostatic repulsion between C60 fullerenes. We would like to note that the research presented in the current paper aims to clarify biological effects of the proposed treatment strategy. Therefore, we would like to address the concern of the Reviewer with the description of our experimental work-flow and respective experimental data (not included in the manuscript, partially published before), that should hopefully weaken the concern. Dynamic Light Scattering (DLS), uv-vis spectroscopy and high-performance liquid chromatography–mass spectrometry measurements were done for each newly synthesized complex as well as during its storage with monthly repetition. The stability (ζ-potential value) and size distribution (hydrodynamic diameter) of nanocomplexes as described in [45, 55] was systematically checked and shown to be practically unchanged after 6 months of storage in physiological saline solution. The biological applicability was proven in our recent study where C60-Dox 1:1 and 2:1 nanocomplexes were shown to remain its average hydrodynamic diameter around 135 ± 14 nm and 134 ± 16 nm respectively in physiological saline solution (0.9 % NaCl) up to 6 months [55]. To estimate the stability in cell culture medium 1 µM C60-Dox nanocomplexes were incubated at 370C for 72 h in RPMI supplemented with 10 % FBS (standard cell culture media for CCRF-CEM cell line). The detected stability of the maximum (≤ 140 nm) indicated that there was no additional aggregation of the C60-Dox nanocomplexes and no diffusion of Doc from the nanocomplexes during a prolonged incubation in FBS-supplemented cell culture medium which confirmed their suitability for in vitro study [55]. We believe that the manuscript would not gain substantial new information from the additional experimental data that more specifically aim on storage stability of complexes in physiological saline solution or cell culture media, moreover those data have been already published before. We added the respective references and the detailed description of the stability, binding and diffusion of nanocomplexes in the “Methods: C60 and C60-Dox nanocomplex synthesis” part. Thank you for your time!

Round 2
Reviewer 1 Report
The manuscript has been revised well. I think this manuscript will be acceptable.
Author Response
Dear Reviewer,
On behalf of all co-authors, I am thanking you for the constructive second revision of our manuscript entitled “Synergy of Chemo- and Photodynamic Therapies with C60 Fullerene-Doxorubicin Nanocomplex” (Manuscript ID: nanomaterials-618446).
The authors are grateful to you for your help and time!
the thoughtful suggestions that helped to enhance the quality of this manuscript. Based on the received comments, we have added required experimental data to the manuscript. All new changes (as compared with the version submitted to the first revision) were highlighted in red in the manuscript. A point-by-point response with further action from our side in a response to the review is included below.
Reviewer 2 Report
The authors rebuttal document does not address a critical point (point 4) raised by Reviewer 2 in the initial review, i.e release of free Dox from C60 in the various experiments. The key paper addressing the stability of C60-Dox in solution is the work by, Evstigneev et al., their reference 48, which is highlighted in red in the revised manuscript. In the present study, the investigators adsorb Dox onto the surface of C60 and isolate the C60-Dox particles. Since the Dox molecules are not covalently attached to the surface of C60, Dox is allowed to diffuse away from C60 when the C60-Dox particles are introduce into solutions in the various experiments. The detailed paper by Evstigneev et al., determined the equilibrium constant of Dox toward C60, which under conditions of the Evstigneev experiments, is actually a complicated cluster of C60 monomers. Evstigneev et al used two different models to interpret the optical titration curves of Dox binding to C60, one model resulting in an equilibrium constant of K = 46000 M^-1 and a second model giving K = 61900 M^-1. The authors of this manuscript erroneously state a value of K = 6000 M^-1 (in red type) which is neither of the above values. The authors say in text that the cytotoxicity studies with C60-Dox involving cells were done in 25 ml flasks but the volume of the culture medium is not stated. When C60-Dox is added to these flasks in the cytotoxicity studies, free Dox will be immediately released from the surface of C60 until the system reaches equilibrium. Exactly, how much Dox is on C60 and how much is free in solution will depend on the total volume of the solution, the length of time that the C60-Dox particles were in the solution, and of course K. Since the total time for the cell experiments is 48 h, and disassociation kinetics for Pi-Pi interactions are generally fast, some free drug must be present in solution. The important control experiment, which is missing in this manuscript, is to measure the rate of release of Dox from C60 using conditions that mimic those that are used in the various experiments. Such a plot would be critical for the potential development of C60-Dox into a practical drug. Without additional experiments, this manuscript is not up to the standards set by the journal.
Author Response
Dear Reviewer,
On behalf of all co-authors, I am thanking you for the constructive second revision of our manuscript entitled “Synergy of Chemo- and Photodynamic Therapies with C60 Fullerene-Doxorubicin Nanocomplex” (Manuscript ID: nanomaterials-618446).
The authors are grateful to you for the thoughtful suggestion that help to enhance the quality of this manuscript. Based on the received comments, we have added required experimental data to the manuscript. All new changes (as compared with the version submitted to the first revision) were highlighted in red in the manuscript. A point-by-point response with further action from our side in a response to the review is included below.
Response to Reviewer 2 Comments
The authors rebuttal document does not address a critical point (point 4) raised by Reviewer 2 in the initial review, i.e release of free Dox from C60 in the various experiments.
Point 1: The key paper addressing the stability of C60-Dox in solution is the work by, Evstigneev et al., their reference 48, which is highlighted in red in the revised manuscript. In the present study, the investigators adsorb Dox onto the surface of C60 and isolate the C60-Dox particles. Since the Dox molecules are not covalently attached to the surface of C60, Dox is allowed to diffuse away from C60 when the C60-Dox particles are introduce into solutions in the various experiments. The detailed paper by Evstigneev et al., determined the equilibrium constant of Dox toward C60, which under conditions of the Evstigneev experiments, is actually a complicated cluster of C60 monomers. Evstigneev et al used two different models to interpret the optical titration curves of Dox binding to C60, one model resulting in an equilibrium constant of K = 46000 M^-1 and a second model giving K = 61900 M^-1. The authors of this manuscript erroneously state a value of K = 6000 M^-1 (in red type) which is neither of the above values.
Response 1: Huge thanks to Reviewer for the sharp eye! It was indeed a misfortunate misspelling. K was meant 60000 M^-1 [47]. The proper value is included in the text of the manuscript. The authors are very grateful to Reviewer for pointing on it!
Point 2: The authors say in text that the cytotoxicity studies with C60-Dox involving cells were done in 25 ml flasks but the volume of the culture medium is not stated.
Response 2: Thank you for the note! We added information about volume of RPMI medium (5 mL) used for cell culture in 25 cm2 flasks. We would like to highlight that 25 cm2 flasks were used only for cell culturing (without any treatment).
Point 3: When C60-Dox is added to these flasks in the cytotoxicity studies, free Dox will be immediately released from the surface of C60 until the system reaches equilibrium. Exactly, how much Dox is on C60 and how much is free in solution will depend on the total volume of the solution, the length of time that the C60-Dox particles were in the solution, and of course K. Since the total time for the cell experiments is 48 h, and disassociation kinetics for Pi-Pi interactions are generally fast, some free drug must be present in solution. The important control experiment, which is missing in this manuscript, is to measure the rate of release of Dox from C60 using conditions that mimic those that are used in the various experiments. Such a plot would be critical for the potential development of C60-Dox into a practical drug. Without additional experiments, this manuscript is not up to the standards set by the journal.
Response 3: Thank you for raising important point! Reviewer made a well taken argument also of interest for any reader of the manuscript. To address the concern of the Reviewer we conducted additional study and added the obtained data to the manuscript to extend characterization of the nanocomplexes.
We would like to clarify that cells were incubated in the presence of C60-Dox during 24 h during all analyses, included in the current manuscript (2. Materials and Methods, 2.7 PDT in vitro and cell viability assay). Taking into account the time given for additional experiment for major revision (7 days), we have conducted Dox release study in cell culture medium RPMI during 24 h three times.
For that C60-Dox nanocomplexes were incubated in RPMI up to 24 h under the identical conditions adopted from cell-based experiments (2 mL, 37 0C). For sample purification from a released free drug, 500 µL of each sample was filtered with the centrifugal filter devices Amicon Ultra-0.5 3 K (Sigma-Aldrich Co., St-Louis, USA) according to manufacturer’s instructions: 14000 g, 15 min for filtration; 1000 g, 2 min for recovery (reverse spin upside down in a new centrifuge tube).
According to our previous results [46] we can conclude that the binding of Dox molecules mainly occurs with C60 clusters in aqueous solution. Theoretical analysis of possible hydrated C60 clusters had shown, that the smallest stable spherical-like C60 cluster consists of 13 C60 molecules and has a diameter 3.36 nm (accounting for a molecular diameter of the water molecule) [Prylutskyy, Yu. I.; Durov, S.S.; Bulavin, L.A. et al. Structure and thermophysical properties of fullerene C60 aqueous solutions. Int J Thermophys 2001 22, 943]. Since up to three Dox molecules can be immobilized on the surface of one C60 molecule in an aqueous medium [48,49], then the aforementioned nanocluster and larger ones are capable of accepting more than 20 Dox molecules. In this case, a molecular weight of possible C60-Dox nanocomplexes exceeds 20000 Da, that is bigger than a cutoff the centrifugal filter device Amicon Ultra-0.5 3 K in 3000 nominal molecular weight.
The content of the filter device was subjected to the optical analysis. C60-Dox nanocomplexes samples (50 μL) were placed into 384-well plate Sarstedt and fluorescence intensities were measured with a multimode microplate spectrometer Tecan Infinite M200 Pro at the following parameters: λex = 470 nm, λem = 595 nm, number of flashes per well: 25, integration time: 20 µs. The obtained data were normalized with the RPMI control and expressed as % of the respective control sample, analyzed at 0 h.
The content of 1:1 and 2:1 C60-Dox nanocomplexes after incubation in RPMI medium for 24 h was assessed to account 81.50 ± 5.03% and 83.83 ± 5.47% correspondingly of respective 0 h control (Fig. 8).
Thank you for your time and help!
in vitro

Round 3
Reviewer 2 Report
In lines 290-292 of the manuscript, the authors say, “For cytotoxicity studies free C60, Dox or C60-Dox nanocomplexes in 50, 150 and 450 nM Dox equivalent concentrations were added to the leukemic cells at time point 0 h. At 24 h cells were illuminated with 405 nm LED and after additional 24 h of incubation cell viability was estimated with MTT assay.” This statement indicates that the total length of time that the nanoparticles were in solution with the cells was 48 h and not 24 h as stated in the rebuttal comments to the reviewer. If the total length of time that the cells were exposed to the particles was actually 24 h, the above statement needs to be corrected.
The authors say that the total solution volume for the cytotoxicity experiments was 5 ml (now stated in the revised manuscript) but in the important control experiment (now in Appendix A as Figure 8) they use a total solution volume of 2 ml and the concentrations of the DOX-C60 complexes are not stated. Figure 8 shows that when the solution volume is 2 ml, about 17% of Dox comes off C60 in about 10 h. Basic principles governing equilibrium mandate that if the solution volume were increased to 5 ml, more Dox (greater than just 17%) would be released from C60. It would be appropriate to redo this experiment using a volume of 5 ml and place the results (the rate of release curve) near the beginning of the manuscript and not at the end of the work as an Appendix. Otherwise, the reader may get the impression that drug loaded C60 particles are stable entities in solution which is not the case.
Author Response
Dear Reviewer,
On behalf of all co-authors, I am thanking you for the constructive third revision of our manuscript entitled “Synergy of Chemo- and Photodynamic Therapies with C60 Fullerene-Doxorubicin Nanocomplex” (Manuscript ID: nanomaterials-618446).
The authors are grateful to you for the thoughtful suggestions, that help to enhance the quality of the manuscript. Based on the received comments, we have changed the order of figures and added required information in the manuscript. All new changes (as compared with the version submitted to the second revision) were highlighted in red. A response with further explanation of used experimental set-up in a response to the third review is included below.
Response to Reviewer 2 Comments
Point 1:
In lines 290-292 of the manuscript, the authors say, “For cytotoxicity studies free C60, Dox or C60-Dox nanocomplexes in 50, 150 and 450 nM Dox equivalent concentrations were added to the leukemic cells at time point 0 h. At 24 h cells were illuminated with 405 nm LED and after additional 24 h of incubation cell viability was estimated with MTT assay.” This statement indicates that the total length of time that the nanoparticles were in solution with the cells was 48 h and not 24 h as stated in the rebuttal comments to the reviewer. If the total length of time that the cells were exposed to the particles was actually 24 h, the above statement needs to be corrected.
Response 1:
Thank you for your recommendation! We would like to address the concern of Reviewer with explanation included below and deletion of the referenced sentence.
The detailed description of set-up for the referenced experiments was included in the section: “2.7. PDT in vitro and cell viability assay”.
CCRF-CEM cells were incubated for 24 h in RPMI medium alone to adopt the cells to changed cultivation conditions. Then, medium was changed on a drug-contained medium and cells were incubated for next 24 h. After those 24 h of incubation drug-contained medium was changed on PBS in order to wash cells from unabsorbed nanocomplexes and perform light irradiation. Immediately after irradiation, PBS was replaced with a fresh RPMI medium, which did not contain any studied agent. At 24 h after light exposure cells were assessed with different cell-based techniques as cell viability.
Therefore, it can be concluded that the experiment had an overall duration in 3 days (72 h). However, CCRF-CEM cells were exposed to the free or C60-complaxeted Dox only for 24 h. Therefore, the duration of cells incubation with C60-Dox nanocomplexes in 24 h was adopted for drug-release study.
The sentence to which Reviewer refers was meant as a brief reminder of experimental set-up and introduction to the results to make the manuscript text more confluent for the reader. We have deleted that sentence to avoid possible misleading. We are grateful to Reviewer for pointing on it!
Point 2:
The authors say that the total solution volume for the cytotoxicity experiments was 5 ml (now stated in the revised manuscript) but in the important control experiment (now in Appendix A as Figure 8) they use a total solution volume of 2 ml and the concentrations of the DOX-C60 complexes are not stated. Figure 8 shows that when the solution volume is 2 ml, about 17% of Dox comes off C60 in about 10 h. Basic principles governing equilibrium mandate that if the solution volume were increased to 5 ml, more Dox (greater than just 17%) would be released from C60. It would be appropriate to redo this experiment using a volume of 5 ml and place the results (the rate of release curve) near the beginning of the manuscript and not at the end of the work as an Appendix. Otherwise, the reader may get the impression that drug loaded C60 particles are stable entities in solution which is not the case.
Response 2:
We thank to Reviewer for precise evaluation of the used experimental work-flow. We would like to highlight the referenced differences in volumes with the precise explanation of the used experimental work-flow below.
In order to use cells for any cell-based experiment, cells should be maintained in culture continuously. For that, cells were grown under common culturing conditions, which were described in the “2.4. Cell culture” part – CCRF-CEM cell line was constantly maintained in 5 mL RPMI in 25 cm2 flask. Cell density was maintained between 2×105 and 2×106 viable cells/mL according to American Type Culture Collection (ATCC) recommendations.
“2.7. PDT in vitro and cell viability assay” described the methodology for studied treatment of CCRF-CEM cells.
When CCRF-CEM cells were grown in required quantity, cells were transferred in a fresh medium by centrifugation with subsequent resuspension. Cells were diluted to the required concentration (stated in each cell-based method description) and seeded in the well plates in 2 mL, where cells were subjected to the treatment with free and C60-complaxeted Dox.
Therefore, it can be concluded that 5 mL volume of RPMI was used only for cell culture maintenance. Cells, grown in 5 mL RPMI, were not treated with C60-Dox nanocomplexes.
We have also rearranged the order of the Figures and stated the concentration of C60-Dox nanocomplexes in drug release description following the recommendations of Reviewer. We thank to Reviewer for pointing on it!
Thank you for your time and help!

Round 4
Reviewer 2 Report
This manuscript is now up to the standards set by the journal.